# Proenkephalin as a Novel Prognostic Marker in Heart Failure Patients: A Systematic Review and Meta-Analysis

**DOI:** 10.3390/ijms24054887

**Published:** 2023-03-03

**Authors:** Noppachai Siranart, Khamik Laohasurayotin, Tanattida Phanthong, Walit Sowalertrat, Aekarach Ariyachaipanich, Ronpichai Chokesuwattanaskul

**Affiliations:** 1Faculty of Medicine, Chulalongkorn University, King Chulalongkorn Memorial Hospital, Thai Red Cross Society, Bangkok 13300, Thailand; 2Division of Cardiology, Center of Excellence in Arrhythmia Research, Department of Medicine, Faculty of Medicine, Chulalongkorn University, King Chulalongkorn Memorial Hospital, Thai Red Cross Society, Bangkok 13300, Thailand

**Keywords:** proenkephalin, heart failure, prognostic marker, mortality, rehospitalization

## Abstract

Over the last several years, the use of biomarkers in the diagnosis of patients with heart failure (HF) has skyrocketed. Natriuretic peptides are currently the most widely used biomarker in the diagnosis and prognosis of individuals with HF. Proenkephalin (PENK) activates delta-opioid receptors in cardiac tissue, resulting in a decreased myocardial contractility and heart rate. However, the goal of this meta-analysis is to evaluate the association between the PENK level at the time of admission and prognosis in patients with HF, such as all-cause mortality, rehospitalization, and decreasing renal function. High PENK levels have been associated with a worsened prognosis in patients with HF.

## 1. Introduction

Nearly 64.3 million individuals worldwide are living with heart failure [1]. Heart failure (HF) is increasingly recognized as a major clinical and public health issue in the global population. The prevalence of known cases of HF is estimated to be 1% to 2% of the adult population in developed countries [2], and it is expected to rise 46% by 2030 in the US population [3].

Over the past several decades, the subject of biomarkers has garnered extensive research in the routine care and management of patients with heart failure. Circulating cardiac biomarkers are significant for the diagnosis, management, and prognosis of cardiac disorders, particularly heart failure. The current approved biomarkers for HF are natriuretic peptides such as B-type natriuretic peptide (BNP) and the N-terminal fragment of its prohormone (NT-proBNP) [4]. Furthermore, a novel biomarker known as proenkephalin (PENK) has been studied separately and in combination in the context of heart failure [5]; however, there has been no substantial improvement in cardiovascular mortality [6].

Several studies have been evaluated, ranging from the primary molecular mechanisms to clinical studies of PENK that suspect a link between increasing PENK and worsening cardiac function. Various aspects [5,7,8,9,10,11] of this biomarker, including all-cause mortality, rehospitalization, and decreasing renal function, are becoming more acknowledged issues in general populations around the world. However, the relationship of its predictive significance in heart failure patients remains unclear. The purpose of this systematic review and meta-analysis is to determine the association between plasma PENK levels and their prognostic values in heart failure patients.

## 2. Materials and Methods

### 2.1. Literature Review and Search Strategy 

A systematic literature search from the MEDLINE (via PubMed), EMBASE (via Scopus), and Cochrane Database of Systematic Reviews databases was conducted for studies up until September 2021. The purpose is to identify studies that measured the associations between plasma PENK levels and prognostic values such as all-cause mortality, rehospitalization, and worsening renal function in HF patients. Two investigators (N.S. and K.L.) independently gathered the systematic literature review using a search strategy that included the terms ‘proenkephalin’ and ‘heart failure,’ as stated in the Appendix A. There were no language restrictions. The reference lists of the recognized studies were manually searched for relevant research as well. This systematic review and this meta-analysis follow the Meta-analyses Of Observational Studies in Epidemiology (MOOSE) standards. In addition, it was conducted in accordance with the Preferred Reporting Items for Systematic Reviews, Meta-Analyses (PRISMA) statement. 

### 2.2. Selection Criteria

The qualified studies must be cross-sectional, case-control, or cohort studies that assess the relationship between plasma PENK levels and prognostic values in HF patients. Case reports and case series were excluded from the study. They must give the estimates of effects including the odds ratios (ORs), relative risks (RRs), or hazard ratios (HRs) with 95% confidence intervals (CI). The inclusion of studies was not restricted by sample size or population ethnicity. Any disagreements concerning study choices were settled through collaborative conversation. As indicated in Table 1, the Newcastle–Ottawa quality assessment scale and the modified Newcastle–Ottawa scale were used to assess the quality of the study for the case-control study, the result of interest for the cohort study, and the cross-sectional study, respectively [12,13]. The evaluation was revealed in three domains: study group selection (S) for four items, group comparability (C) for two items, and exposure and outcome (O) for three items. The bias assessment results were displayed as a number, with a maximum of 4, 2, and 3 for the S, C, and O domains, respectively. A higher number implies a higher quality. 

### 2.3. Data Abstraction

The following information was collected from each study using a structured data record form: title, year of the study, name of the first author, publication year, country where the study was conducted, patient baseline characteristics, demographic data, and method used to identify heart failure patients and measure the plasma PENK level. In addition to the previously mentioned data collection, classifications of the PENK level in each group and definitions of all-cause mortality, rehospitalization, and worsening renal function were also collected. Adjust effect estimates using 95 percent confidence intervals and factors that were modified in the multivariable analysis were used in the study.

### 2.4. Definition of a High Plasma PENK Level

According to the literature review [5,7,8,9,10,11] (Table 1), subjects were subdivided into smaller groups based on plasma PENK levels measured using a chemiluminometric sandwich immunoassay at the time of arrival. The plasma PENK cut-off level varies between research, although it may be inferred that the high plasma PENK level refers to the measured level above 67–83 pmol/L.

### 2.5. Definition of Worsening Renal Function

According to the literature review, worsening renal function is defined as a decrease of more than 25% in the estimated glomerular filtration rate (eGFR) compared to baseline [10] or an increase of more than 0.3 ng/dL or of 50% of the level within 48 h of admission compared to the admission value [7,11].

### 2.6. Statistical Analysis

The R program was used to analyze the obtained data. Adjusted point estimates from each study were calculated using DerSimonian and Laird’s generic inverse variance technique, which assigned a weight to each study based on its variance [14]. A random-effect model was used to separate the HR for all-cause mortality, the HR for rehospitalization, and the OR for worsening renal function among patients with high and normal levels of PENK. The Cochran-Q test was used to examine and quantify the prevalence heterogeneity between studies. If there was heterogeneity (*p* < 0.1 or I^2^ > 25%), the DerSimonian and Laird technique was used; otherwise, an inverse-variance fixed-effect model was utilized [15]. Subsequently, meta-regression and subgroup analysis were used to identify sources of heterogeneity, such as clinical and methodological differences. The Egger test is used to determine whether there is publication bias [16].

### 2.7. Sensitivity Analysis

The sensitivity analysis is computed using a mixed-effect model. The moderator test was statistically insignificant (*p* = 0.11), indicating that the publication year had no effect on the effect size.

### 2.8. Evaluation of Publication Bias

Due to the limited research, the power of the test is too low to distinguish between chance and actual asymmetry; hence, a funnel plot was not produced [17]. For the correlations of the PENK level with all-cause mortality, Egger’s regression asymmetry revealed no publication bias, with *p* = 0.84.

## 3. Results

A total of 119 identified publications were included after removing duplicates using our search method. In addition, 96 studies were excluded from the study following abstract screening, since they were case reports, case series, review articles, in vitro studies, animal studies, or interventional studies. The complete-length review with no extra studies from the reference lists left 23 studies for full texts. The remaining 17 studies were later excluded due to a failure to report the outcome of interest and a lack of an exact plasma PENK cut-off level. As a result, the final analysis included six observational cohort studies [5,7,8,9,10,11] involving 6929 individuals with either acute or chronic HF. Figure 1 depicts the literature review’s inclusion and exclusion process.

### 3.1. Association between a High Plasma PENK Level and All-Cause Mortality

Six studies [5,7,8,9,10,11] evaluated the association between a high plasma PENK level and all-cause mortality, as shown in Table 1. With pooled HRs of 1.72 (95% CI, 1.62–1.84, I^2^ = 84%), a high level of PENK was substantially related with a greater incidence of all-cause death. Figure 2A depicts the analyzed data.

### 3.2. Association between a High Plasma PENK Level and Rehospitalization

Two studies [9,10] investigated the relationship between high plasma PENK levels and rehospitalization (Table 1). A pooled HR of 1.51 (95% CI, 1.38–1.65, I^2^ = 0%; Figure 2B) was shown to be significant between high plasma PENK levels and rehospitalization. 

### 3.3. Association between a High Plasma PENK Level and Worsening Renal Function

Three studies [7,10,11] examined the relationship between elevated plasma PENK levels and deteriorating renal function (Table 1). High plasma PENK levels were associated with deteriorating renal function, with a pooled OR of 1.57 (95% CI, 1.36–1.82, I^2^ = 0%; Figure 2C).

## 4. Discussion

Apart from HF prognoses, the studies revealed a significant correlation between patient variables (e.g., age, body mass index, hypertension, diabetes) and plasma PENK levels. Furthermore, plasma PENK levels are correlated with numerous laboratory findings (serum creatinine level, estimated glomerular filtration rate) and imaging data (ejection fraction, echocardiographic index of diastolic dysfunction E/e’). 

In this meta-analysis, we observed that plasma proenkephalin levels were significantly associated with HF outcomes such as all-cause mortality, rehospitalization, and worsening renal function. When compared to patients with normal plasma PENK levels, those with high plasma levels had a 72 percent increased risk of all-cause mortality and a 51 percent increased risk of rehospitalization. Furthermore, there is a 1.57-fold increase in the likelihood of an incident worsening renal function in HF.

In the past ten years, there have been an increasing number of studies describing the mechanism and clinical significance of PENK in patients with renal or cardiac failure. Proenkephalin (PENK), an endogenous opioid peptide present in the central nervous system and the autonomic nervous system in numerous organs including the heart, is implicated in ischemia preconditioning via the activation of delta-opioid receptors in heart tissue. This opens mitochondrial potassium-ATP channels, which are prevalent in heart mitochondria and prevent cytosolic calcium excess. As a result, the myocardial contractility, blood pressure, and heart rate are lowered [18]. Enkephalins are hypothesized to have a regulatory effect on the function in the kidney, which has the second highest delta-opioid receptor density after the central nervous system. This causes diuresis with natriuresis and the blocking of the antidiuretic hormone. PENK-induced renal function changes result in increased renal blood flow and urine output [18]. PENK, rather than enkephalins, has been utilized as a biomarker for kidney function since it has a longer half-life, and its levels are unaffected by age or sex [19]. PENK has been used to assess kidney function in a variety of settings, including healthy people, post-renal transplant, post-cardiac surgery, and chronic kidney disease. 

The effects of enkephalins on renal blood flow and urine output might cause the production of PENK as a counterregulatory response to renal failure. Without affecting the heart rate or blood pressure, activating enkephalin receptors causes diuretic and natriuretic responses [18,20]. Alternatively, this might lead to decreased renal perfusion due to the cardio-depressive reaction to PENK. Chronic renal hypoxia, renal vein congestion due to poorer HF, and a subsequent increase in tubular damage combined may explain the relationship between PENK and tubular damage. Furthermore, PENK expression is increased in response to oxidative stress conditions, and both renal and HF are linked to a prooxidant state, especially in tubular cells. However, prior research that included chronic and acute HF found no correlation between PENK and indicators of tubular injury [10,21,22].

PENK has also been utilized as a predictive marker in a variety of disorders, including acute myocardial infarction, sepsis, acute ischemic stroke, and diabetes. Nonetheless, its usefulness as a prognostic marker in heart failure patients remains inconclusive. A new investigation stated that the PENK level in acute HF patients might represent their heart and kidney condition; however, after adjustment for other renal biomarkers, including serum creatinine, the link was no longer independent. The statistical outcomes in the recent study supported the conclusion that the baseline plasma PENK and baseline serum creatinine had a high correlation; additionally, a linear association between the plasma PENK and plasma NGAL was demonstrated [9]. 

The prognostic value of plasma PENK for type 1 cardiorenal syndrome (CRS-1) was explored in a study by Zhao et al. Patients with acute decompensated heart failure (ADHF) frequently experience CRS-1, which is described as a fast deterioration of cardiac function followed by acute kidney injury (AKI). The study found that, in hospitalized patients with ADHF, plasma PENK was a strong independent predictor for CRS-1. Furthermore, as plasma PENK increased, the prevalence of CRS-1 increased considerably. Again, in ADHF patients, the plasma PENK was an independent risk factor for HF readmission and all-cause mortality 90 days after discharge [23].

Several indicators have sparked increased interest in the diagnosis, therapeutics, and prognosis of heart failure patients in recent years. Natriuretic peptides (NPs)—notably, B-type natriuretic peptide (BNP) and the N-terminal portion of its prohormone (NT-proBNP)—are the most widely acknowledged biomarkers for heart failure patients. Because of their high negative predictive value, NPs are commonly used as initial diagnostic tests to rule out the diagnosis of HF. Furthermore, in heart failure patients, NPs may lead to the decision to further explore cardiac conditions or predict the prognosis [24]. Nevertheless, the use of NPs to direct therapy in patients with heart failure is still controversial. In comparison to the standard care group, the NT-proBNP-guided method in HF patients did not significantly reduce cardiovascular mortality or the time to initial hospitalization, according to a study by Felker et al. Thus, it is encouraged to conduct more clinical trials examining the prognosis of heart failure patients receiving PENK-guided treatment [6]. Novel biomarkers such as PENK [5], ST-2, GDF-15, Pentraxin-3, Galectin-3, and Osteopontin [25] have recently emerged as potential prognostic indicators in heart failure patients. PENK appears promising for predicting prognoses for cardiac and renal functions, which will decrease as the heart failure worsens. As a result, PENK has the potential to be yet another clinical biomarker used in the prognostic and diagnostic evaluation of heart failure patients. Thus, it is encouraged to conduct more clinical trials examining the prognosis of heart failure patients receiving PENK-guided treatment. 

Plasma PENK was measured by the chemiluminometric sandwich immunoassay technique, which researchers and clinicians have widely used in recent years. Two particular monoclonal antibodies directed against PENK 129–144 (anti-PENK 129–144 mAb) and PENK 152–159 (anti-PENK 152–159 mAb) are used in the measurement of plasma PENK. Samples (50 microliters plasma) were immobilized by the capture antibody (2 micrograms coated on polystyrene tubes). The detector antibody was labeled with methylacridinium ester, and bound chemiluminescence was measured. The results were contrasted with the predicted and observed concentrations of PENK. According to previous research findings, the assay technique successfully measured PENK in human specimens, defined as measuring within 15% of predicted quantities. The outcomes demonstrated a substantial correlation between the assay’s measurement of plasma PENK levels and GFR. Age, sex, and transplant status were significant predictors of GFR in a multivariate model, and 49% of the projected GFR values fell within 30% of the GFR [26].

Emmens et al. [10] discovered that increased plasma PENK levels are related with worsening renal function in their investigation. One probable explanation is that PENK can freely filter through the glomerulus; hence, plasma PENK levels rise when the estimated glomerular filtration rate falls (eGFR). In response to poor renal function, PENK is produced to improve the renal blood flow and promote urine output by activating delta-opioid receptors in the kidney. The use of PENK to predict prognosis in acute myocardial infarction implies that the opioid system is involved in depressive effects on cardiac and renal function. Additionally, Ng et al. discovered that higher PENK concentrations were causally associated with a higher likelihood of developing HF following myocardial infarction [27]. Based on the findings and reviews, we may conclude that the opioid system could be involved in the worsening of cardiovascular functions in heart failure. However, various complicating factors, such as patient characteristics (e.g., age, gender), underlying diseases (e.g., DM, HT), and medications, might raise PENK levels (e.g., ACEI, beta-blockers), resulting in abnormally high PENK levels. 

Numerous studies on the connection between the kidney and the heart [28,29] have been conducted during the last five years. Both organs frequently exhibit co-occurring dysfunction, and cardiac causes can induce renal damage or vice versa. Moreover, there may be shared pathways that affect both cardiac and renal function. Mechanistic evidence supports the opioid system, represented by PENK, as a shared route affecting the heart and kidney. However, only prospective intervention studies can show a causal association. Therefore, it must be determined if the opioid system is causally responsible for the onset of cardiorenal failure. To our knowledge, the therapeutic administration of PENK regarding circulatory effects in HF has not been investigated outside of its usage as an analgesic in experimental animal models.

Additionally, it is speculated that the drug used to treat heart failure may impact the body’s level of circulating PENK—specifically, the angiotensin receptor blocker/neprilysin inhibitors (ARNi), sacubitril, and valsartan. This effect may assist in explaining why ARNi can influence the PENK level, as enkephalins are a substrate of neprilysin [10] and can improve heart failure outcomes.

However, a contradictory study showed the correlation of PENK and the pro-BNP level at a 2-year follow-up following the start of ARNi treatment. Thus, the PENK level is still related to the disease progression [30].

Our study identified a connection between plasma proenkephalin levels and numerous outcomes in heart failure patients, including all-cause death, rehospitalization, and decreasing renal function. These encouraging findings may pave the way for additional research into this biomarker. Furthermore, the findings of this study may encourage the use of proenkephalin as a predictive marker in patients with heart failure in current clinical practices.

There are several limitations in this study. First, the number of studies is limited; as a result, additional research may be required to strengthen the findings. Second, because this is a meta-analysis of observational research, this study can only show relationships between plasma PENK levels and prognosis in heart failure patients, not a causative relationship. Next, in our meta-analysis, there was statistically significant heterogeneity between trials. This could be because of the patient’s gender, age, underlying condition, current medications, and other laboratory results. Furthermore, because the cut-off values in the studies differ, the precise definition of a high plasma PENK level remains unknown. As a result, future research should clearly define the high plasma PENK level threshold. Moreover, existing studies have primarily focused on American and European patients; thus, research on other ethnic groups should be encouraged because plasma PENK levels may change between ethnic groups. Finally, one of the significant limitations that is worth mentioning is the heterogeneity in the enrolled studied population. It also does not distinguish between patients with preserved ejection fraction and those with decreased ejection fraction, which may influence the outcome because of the different pathophysiology, as preserved ejection fraction heart failure may not involve worsening cardiac function, as in high output failure. Most of the included studies combined both subtypes during enrollment. Therefore, a separate analysis between HFrEF and HFpEF was not possible. More research on specific types of patients is thus recommended.

## 5. Conclusions

Our study demonstrated associations of the plasma PENK level and HF prognoses, which are all-cause mortality, rehospitalization, and worsening renal function. These findings could suggest that the opioid system is implicated in the worsening of cardiovascular functioning in heart failure. We encourage additional research into this biomarker element and look forward to the outcomes of prospective randomized trials investigating this issue. PENK level-guided therapy may be a viable option for routine heart failure treatment.

## Figures and Tables

**Figure 1 ijms-24-04887-f001:**
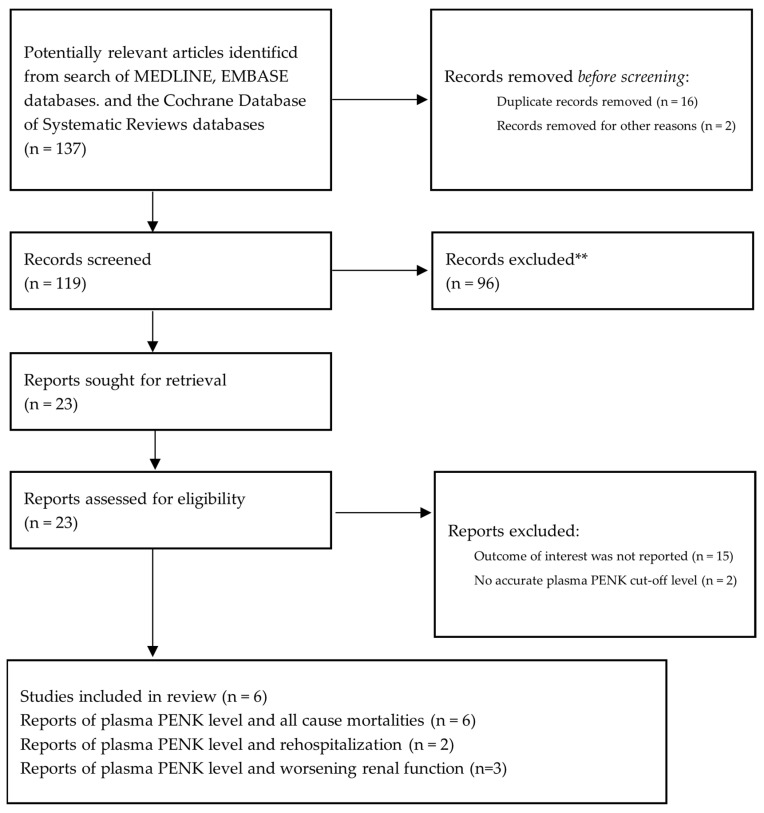
PRISMA flow of our search methodology. ** Records excluded since they were case reports, case series, review articles, in vitro studies, animal studies, or interventional studies.

**Figure 2 ijms-24-04887-f002:**
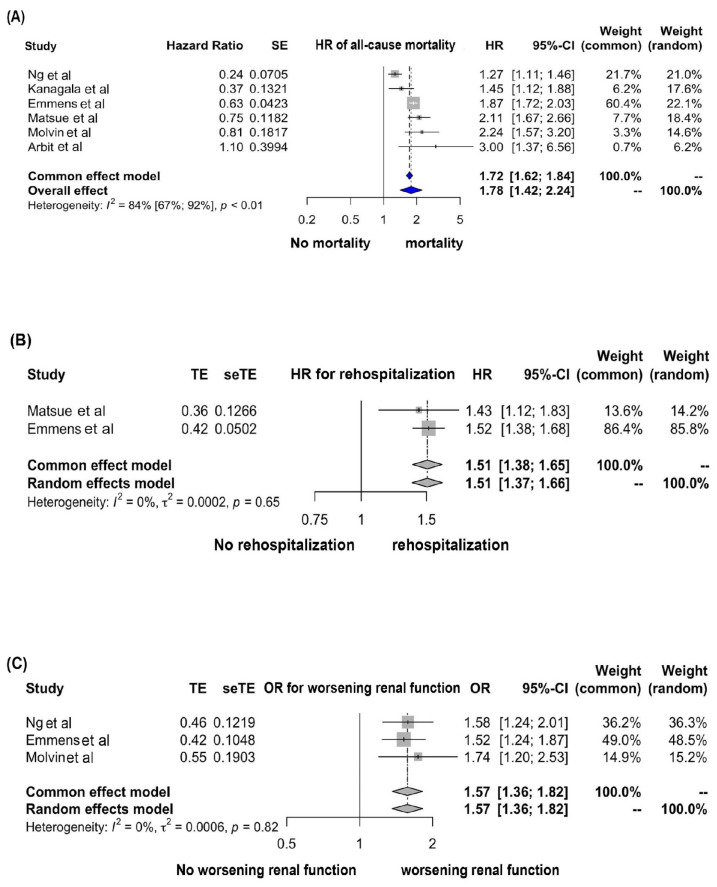
Forest plots of the included studies evaluating the associations of plasma PENK levels at the time of admission and their predictive prognosis including all-cause mortality [5,7,8,9,10,11] (**A**), rehospitalization [9,10] (**B**), and worsening renal function [7,10,11] (**C**) in patients with heart failure by using a random-effect model. Square data markers show hazard ratios (HRs) and odds ratios (ORs); horizontal lines show the 95% confidence intervals (CI), with the marker size indicating the statistical weight of the study using random-effects meta-analysis. A diamond marker shows the overall HRs or ORs and the 95% CI for the outcome of interest.

**Table 1 ijms-24-04887-t001:** Main characteristics of the studies included in the meta-analysis of the associations of plasma PENK levels at the time of admission and their predictive prognosis including all-cause mortality, rehospitalization, and worsening renal function.

	Kanagala et al. [5]	Ng et al. [7]	Arbit et al. [8]	Matsue et al. [9]	Emmens et al. [10]	Molvin et al. [11]
Year	2019	2017	2016	2016	2019	2019
Country	United Kingdom, Switzerland	United Kingdom, France, Switzerland	USA	Netherlands, Germany, Italy, United Kingdom, Poland, USA	Scotland, United Kingdom	Sweden, Italy
Study design	Cohort study	Cohort study	Cohort study	Cohort study	Cohort study	Cohort study
Population	Patients with either new onset or worsening HF	Patients with acute HF who presented with acute dyspnea	Patients with HF who were referred for an echocardiogram	Patients with acute HF with renal function impairment	Patients with either new onset or worsening HF	Patients with either new onset or worsening HF
Settings (OPD/IPD)	Both	Both	OPD	Both	Both	Both
n	522	1908	200	1589	2180	530
Mean age ± SD	76.13 ± 10.73	75.66 ± 11.74	63.7 ± 12.5	71 ± 11	69 ± 12	76.4 ± 10.7
Time to follow-up (months)	24	12	48	6	21	12
Male sex (%)	48.5	62.2	97.5	66	73.2	60.2
CAD	171	769	N/A	N/A	957	N/A
HT	425	1330	143	1272	228	N/A
DM	189	609	49	733	703	194
Loop diuretics (%)	N/A	65.6	N/A	N/A	99.5	82.6
Beta-blocker (%)	62.9	53.9	55.5	75	83.1	71.7
ACEI/ARB (%)	74.4	61.6	34.5	N/A	71.7	67.9
Plasma PENK measurement	chemiluminometric sandwich immunoassay	chemiluminometric sandwich immunoassay	chemiluminometric sandwich immunoassay	chemiluminometric sandwich immunoassay	chemiluminometric sandwich immunoassay	chemiluminometric sandwich immunoassay
PENK cut-off level (pmol/L)	68.2	67	67.5	83	69.4	N/A
All-cause mortality						
Hazard ratio for all-cause mortality	1.45 (1.12–1.88)	1.27 (1.1–1.45)	3 (1.4–6.7)	2.11 (1.68–2.67)	1.87 (1.72–2.03)	2.24 (1.57–3.2)
Counfounder adjusted	Age, sex, SBP, HR, NYHA class, DM, HT, previous HF, IHD, AF, plasma urea, Cr, sodium, Hb, natriuretic peptide	Age, sex, SBP, HR, renal failure, DM, HF, HT, IHD, eGFR, BUN, NP levels	N/A	Age, sex, SBP, peripheral edema, previous HF rehospitalization, serum Na, log BUN, log Cr, albumin	Age, Hb, beta-blocker use, eGFR, BUN, log UACR, log NT-proBNP, log hs-troponin T, log CRP, log GDF-15, log urinary KIM-1, log urinary NGAL, BIOSTAT risk score	Age, sex, DM, SBP, smoking, AF, prior HF, NT-proBNP
Quality assessment	S4C2O2	S3C2O2	S3C1O2	S4C2O2	S4C2O2	S3C2O2
Worsening renal function						
WRF definition	†	Increase in the plasma creatinine level of more than 0.3 ng/dL or of 50% of the level within 48 h of admission compared with the admission value	†	†	Decrease of more than 25% in the estimated glomerular filtration rate (eGFR) compared with the baseline	Increase in the plasma creatinine level of more than 0.3 ng/dL or of 50% of the level within 48 h of admission compared with the admission value
Baseline Cr (mg/dL)	†	1.43 ± 0.74	†	†	1.17 (0.95–1.47)	1.37 ± 0.76
Adjusted odds ratio for WRF	†	1.58 (1.24–2.00)	†	†	1.52 (1.24–1.87)	1.74 (1.2–2.53)
Confounder WRF	†	Age, sex, SBP, HR, renal failure, DM, HF, HT, IHD, eGFR, plasma urea, NP levels	†	†	Plasma NGAL, urinary KIM-1, urinary NGAL, UACR	SBP, ACEI, ARB, beta-blocker, DM, prior HF, Cr, BNP
Quality assessment	†	S2C1O2	†	†	S3C1O2	S2C2O2
Rehospitalization						
Harzard ratio for rehospitalization	‡	‡	‡	1.43 (1.12–1.84)	1.52 (1.38–1.68)	‡
Confounder rehospitalization	‡	‡	‡	Age, sex, SBP, peripheral edema, previous HF rehospitalization, serum Na, log BUN, log Cr, albumin	Age, peripheral edema, SBP, beta-blocker use, previous HF hospitalization, Hb, eGFR, BUN, log UACR, log NT-proBNP, log hs-troponin T, log CRP, log GDF-15, log urinary KIM-1, log urinary NGAL, BIOSTAT risk score	‡
Quality assessment	‡	‡	‡	S4C2O2	S4C2O2	‡

† This study does not report an outcome of worsening renal function. ‡ This study does not report an outcome of HF rehospitalization. ACEI, angiotensin converting enzyme inhibitor; AF, atrial fibrillation; ARB, angiotensin receptor blocker; BNP, brain natriuretic peptide; BMI, body mass index; BUN, blood urea nitrogen; CAD, coronary artery diseases; Cr, creatinine; CRP, C-reactive protein; DM, diabetes mellitus; eGFR, estimated glomerular filtration rate; GDF-15, growth differentiation factor-15; Hb, hemoglobin; HF, heart failure; HR, heart rate; HT, Hypertension; IHD, ischemic heart disease; IPD, in-patient department; KIM-1, kidney injury molecule-1; NGAL, neutrophil gelatinase-associated lipocalin; NP, natriuretic peptide; NT-proBNP, N-terminal pro B-type natriuretic peptide; OPD, out-patient department; SBP, systolic blood pressure; SD, standard deviation; WRF, worsening renal function; UACR, urine albumin-to-creatinine ratio; S, C, O, selection, comparability, and outcome.

## Data Availability

No new data were created or analyzed in this study. Data sharing is not applicable to this article.

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
