# Peer review of "Proenkephalin as a Novel Prognostic Marker in Heart Failure Patients: A Systematic Review and Meta-Analysis"

_ijms, 2023, doi:10.3390/ijms24054887_

Round 1
Reviewer 1 Report
I read with interest the article titled "Proenkephalin as a Novel Prognostic Marker in Heart Failure Patients: A Systematic Review and Meta-Analysis"; In this meta-analysis, the authors sought to analyze the association between PENK level at hospitalization and prognosis in heart failure patients, such as all-cause mortality, rehospitalization, and impaired renal function. They found that high PENK levels are associated with a worsening prognosis in patients with heart failure.
The paper is well written and easily understandable. It is also statistically rigorous.
References are detailed.
I suggest the authors insert a short paragraph on the clinical and technical possibilities of this evidence.
Author Response
Response to Reviewer
Reviewer: 1
COMMENTS FOR THE AUTHOR(S)
I read with interest the article titled "Proenkephalin as a Novel Prognostic Marker in Heart Failure Patients: A Systematic Review and Meta-Analysis"; In this meta-analysis, the authors sought to analyze the association between PENK level at hospitalization and prognosis in heart failure patients, such as all-cause mortality, rehospitalization, and impaired renal function. They found that high PENK levels are associated with a worsening prognosis in patients with heart failure.
The paper is well written and easily understandable. It is also statistically rigorous.
References are detailed.
I suggest the authors insert a short paragraph on the clinical and technical possibilities of this evidence.
Response: Thank you, we appreciated your time and effort to review our manuscript. For the clinical and technical possibilities of this evidence with the research progression in the past 5 years, we have already inserted in Line 186-196, 200-212, 221-226, 262-278, of the manuscript with the following text.
“
The effects of enkephalins on renal blood flow and urine output might cause the production of PENK as a counterregulatory response to renal failure. Without affecting heart rate or blood pressure, activating enkephalin receptors causes diuretic and natriuretic responses. Alternatively, this might lead to decreased renal perfusion due to the cardio-depressive reaction to PENK. Chronic renal hypoxia, renal vein congestion due to poorer HF, and a subsequent increase in tubular damage combined may explain the relationship between PENK and tubular damage. Furthermore, PENK expression is increased in response to oxidative stress conditions, and both renal and HF are linked to a prooxidant state, especially in tubular cells. However, prior research that included chronic and acute HF found no correlation between PENK and indicators of tubular injury.
New Investigation stated that the PENK level in acute HF patients might represent their heart and kidney condition; however, after adjustment for other renal biomarkers, including serum creatinine, the link was no longer independent. The statistical outcomes in the recent study supported the conclusion that baseline plasma PENK and baseline serum creatinine had a high correlation; additionally, a linear association between plasma PENK and plasma NGAL was demonstrated.
The prognostic value of plasma PENK for type 1 cardiorenal syndrome (CRS-1) was explored in a study by Zhao et al. Patients with acute decompensated heart failure (ADHF) frequently experience CRS-1, which is described as a fast deterioration of cardiac function followed by acute kidney injury (AKI). The study found that in hospitalized patients with ADHF, plasma PENK was a strong independent predictor for CRS-1. Furthermore, as plasma PENK increased, the prevalence of CRS-1 increased considerably. Again, in ADHF patients, the plasma PENK was an independent risk factor for HF readmission and all-cause mortality 90 days after discharge.
Nevertheless, the use of NPs to direct therapy in patients with heart failure is still controversial. In comparison to the standard care group, the NT-proBNP-guided method in HF patients did not significantly reduce cardiovascular mortality or the time to initial hospitalization, according to a study by Felker et al. Thus; it is encouraged to conduct more clinical trials examining the prognosis of heart failure patients receiving PENK guided treatment.
Numerous studies on the connection between the kidney and the heart have been conducted during the last five years. Both organs frequently exhibit co-occurring dysfunction, and cardiac causes can induce renal damage or vice versa. Moreover, there may be shared pathways that affect both cardiac and renal function. Mechanistic evidence supports the opioid system, represented by PENK, as a shared route affecting the heart and kidney. However, only prospective intervention studies can show a causal association. Therefore, it must be determined if the opioid system is causally responsible for the onset of cardiorenal failure. To our knowledge, the therapeutic administration of PENK regarding circulatory effects in HF has not been investigated outside of its usage as an analgesic in experimental animal models.
Additionally, it is speculated that the drug used to treat heart failure may impact the body's level of circulating PENK. Specifically, the angiotensin receptor blocker/neprilysin inhibitors (ARNi), sacubitril, and valsartan. This effect may assist in explaining why ARNi can influence PENK level, as enkephalins are a substrate of neprilysin and can improve heart failure outcomes.
However, a contradictory study showed the correlation of PENK and pro-BNP level at a 2-year follow-up following the start of ARNi treatment. Thus, the PENK level is still related to the disease progression.
"

Reviewer 2 Report
See attached file.

Author Response
Response to Reviewer
Reviewer: 2
COMMENTS FOR THE AUTHOR(S)
Summary
The authors performed a systematic review and meta-analysis (MOOSE and PRISMA guided) of the prognostic value of admission plasma proenkephalin (PENK) concentrations in HF (mixture of acute and chronic HF). They identified 6 eligible observational studies (2016-2019 on 6929 patients) for all-cause mortality as endpoint, 2 studies for re -hospitalization and 3 studies for worsening of renal failure as endpoints. PENK has been assayed with a chemiluminometric sandwich immunoassay. The meta-analysis was performed by the DerSimonian - Laird generic inverse variance technique with a random-effect model for total mortality and a fixed-effect model for the other outcomes. The outcomes were significantly associated with plasma PENK: adj pooled HR of 1.72 for total-mortality (important heterogeneity), adj pooled HR of 1.51 for re-hospitalization (no heterogeneity) and adj pooled OR of 1.57 (no heterogeneity) for worsening of renal function.
Comments
- The meta-analysis provides the reader with confirmatory evidence that admission plasma PENK predicts all-cause mortality, re-hospitalization and worsening of the renal function. The novelty is limited as each eligible study significantly predicted the outcomes in its own right. Thus, no surprise that the pooled HRs and OR are The findings on heterogeneity analyses are striking: sqI of 84% for all-cause mortality whereas 0% heterogeneity for the other outcomes. Those data are a call for more studies. The study question is of interest because of the high activation of delta-opioid receptors in the kidney. However, after reading the manuscript the reviewer has also a number of questions and criticisms.
Response: Thank you, we appreciated your time and effort to review our manuscript. We have improved our manuscript regarding your comments.
- You are dealing with the prognostic value of a biomarker. The prognostic value is ideally calculated from prognostic studies which prospectively followed the patients for some time (follow-up time). In the text (lines 59-60) we read that “Qualified studies must be cross-sectional, case-control or cohort studies that assess the relationship between plasma PENK and prognostic values in HF patients”. The authors should clarify that sentence and where possible the median time of follow-up should be specified. It is important as the follow-up time also defines the likelihood to observe some of the hard endpoints.
Response: We added time to follow-up of each study in the table1 to better illustrate the outcome observation.
- Do the authors have a reference on the validation of the chemiluminometric sandwich immunoassay for plasma PENK? It would further strengthen the credibility of the findings.
Response: We described the validation of chemiluminometric sandwich immunoassay for plasma PENK in Line 234-246 the text read.
Plasma PENK was measured by the chemiluminometric sandwich immunoassay technique, which researchers and clinicians have widely used in recent years. Two particular monoclonal antibodies—directed against PENK 129-144 (anti-PENK 129-144 mAb) and PENK 152-159 (anti-PENK 152–159 mAb) are used in the measurement of plasma PENK. Samples (50 microliters plasma) were immobilized by the capture antibody (2 micrograms coated on polystyrene tubes). The detector antibody was labeled with methylacridinium ester, and bound chemiluminescence was measured. The results were contrasted with the predicted and observed concentrations of PENK. According to previous research findings, the assay technique successfully measured PENK in human specimens, defined as measuring within 15% of predicted quantities. The outcomes demonstrated a substantial correlation between the assay's measurement of plasma PENK levels and GFR. Age, sex, and transplant status were significant predictors of GFR in a multivariate model, and 49% of projected GFR values fell within 30% of the GFR.
- The strong heterogeneity for all-cause mortality, the mixture of acute and chronic conditions and of HFpEF and HFrEF makes final conclusions a little bit difficult and uncertain.
Response: We agreed with your comment. We think that the distinction of preserved and reduced ejection fraction heart failure patients and also acute and chronic heart failure should be made in further studies. The limitations [line 297-303] were also revised as well as attached below.
One of the significant limitations that is worth mentioning is heterogeneity in the enrolled studied population. It also does not distinguish between patients with preserved ejection fraction and those with decreased ejection fraction, which may influence the outcome because of different pathophysiology, as preserved ejection fraction heart failure may not involve worsening cardiac function, such as in high output failure. Most included studies combined both subtypes during enrollment. Therefore, a separate analysis between HFrEF and HFpEF was not possible.

Round 2
Reviewer 2 Report
The authors have answered the questions which have been raised by the reviewer. The reviewer does not have further comments.